# WASSERSTEIN ADVERSARIAL REGULARIZATION (WAR) ON LABEL NOISE

## ABSTRACT

Noisy labels often occur in vision datasets, especially when they are obtained from crowdsourcing or Web scraping. We propose a new regularization method, which enables learning robust classifiers in presence of noisy data. To achieve this goal, we propose a new adversarial regularization scheme based on the Wasserstein distance. Using this distance allows taking into account specific relations between classes by leveraging the geometric properties of the labels space. Our Wasserstein Adversarial Regularization (WAR) encodes a selective regularization, which promotes smoothness of the classifier between some classes, while preserving sufficient complexity of the decision boundary between others. We first discuss how and why adversarial regularization can be used in the context of label noise and then show the effectiveness of our method on five datasets corrupted with noisy labels: in both benchmarks and real datasets, WAR outperforms the state-of-the-art competitors.

## 1 INTRODUCTION

Deep neural networks require large amount of accurately annotated training samples to achieve good generalization performances. Unfortunately, annotating large datasets is a challenging and costly task, which is practically impossible to do perfectly for every task at hand. It is then most likely that datasets will contain incorrectly labeled data, which induces noise in those datasets and can hamper learning. This problem is often referred to as *learning with label noise* or *noisy labels*. The probability of facing this problem increases when the dataset contains several fine grained classes that are difficult to distinguish (Schroff et al., 2011; Krause et al., 2016; Dubey et al., 2018). As pointed out in (Zhang et al., 2017), deep convolutional neural networks have huge memorization abilities and can learn very complex functions. That is why training with noisy data labels can lead to poor generalization (Arpit et al., 2017; Wang et al., 2018; Choi et al., 2018). Hence in this paper we propose a method tackling the problem of overfitting on noisy labels, and this without access to a clean validation dataset.

This problem has been considered in recent literature, mainly in three ways. First are *data cleaning methods*: (Vahdat, 2017; Xiao et al., 2015; Li et al., 2017) learn relations between noisy and clean labels before estimating new labels for training. In (Lee et al., 2018), few human verified labels were necessary to detect noisy labels and adapt learning. In (Jiang et al., 2018; Ren et al., 2018), the methods rely either on a curriculum or on meta-gradient updates to re-weight training sets and downweight samples with noisy labels. Second are *Transition probability-based methods*: (Sukhbaatar & Fergus, 2014; Patrini et al., 2017; Hendrycks et al., 2018) estimate a probability for each label to be flipped to another class and use these estimations to build a noise transition matrix. In (Sukhbaatar & Fergus, 2014), the authors add an extra linear layer to the softmax in order to learn the noise transition matrix itself, while (Hendrycks et al., 2018) uses a small set of clean validation data to estimate it. (Patrini et al., 2017) proposes a forward/backward loss correction method, which exploits the noise transition matrix to correct the loss function itself. Third are *regularization-based methods* . In (Reed et al., 2015; Ma et al., 2018), the authors use a mixture between the noisy labels and network predictions. In (Tanaka et al., 2018), the regularization is achieved by alternatively optimizing the network parameters and estimating the true labels while the authors of (Han et al., 2018; Yu et al., 2019) propose peer networks feedbacking each other about predictions for the noisy labels. Song et al. (2019) proposes to replace noisy labels in the mini-batch by the consistent network predictions during training, while Chen et al. (2019) proposes noisy cross-validation to identify samples

that have correct labels. In (Wang et al., 2019; Zhang & Sabuncu, 2018; Ghosh et al., 2017), robust loss functions are proposed to overcome limitations of cross entropy loss function.

In contrast to those works, we propose to regularize predictions in areas of the feature space close to the decision boundary of conflicting classes, therefore mitigating the influence of noisy labels. To do so, we use the adversarial regularization (AR) framework (Goodfellow et al., 2015; Miyato et al., 2018) on the noisy label problem. We use AR to reduce the discrepancy between the prediction of a true input sample and the one obtained by a near-by adversarial sample. To reduce this discrepancy, we use a loss based on the Wasserstein distance computed with respect to a ground cost encoding class similarities. This ground cost provides the flexibility to regularize with different strengths pairs of classes. This strength can depend on semantic relations, classes similarities, or prior knowledge (e.g. on annotators' mistakes). This way, the classifier can discriminate non-similar objects robustly under the presence of noise and class overlap. We name our proposed method *Wasserstein Adversarial Regularization* (WAR). WAR allows incorporating specific knowledge about the potential degree of mixing of classes through a ground cost that can be designed *w.r.t.* the problem at hand. Nevertheless, this knowledge might be unknown or difficult to craft. In this paper, we use distances between word embeddings of the class names to derive a semantic ground cost. Experiments in five datasets (Fashon-MNIST, CIFAR10, CIFAR100 and real life examples on clothing classification and a remote sensing semantic segmentation dataset) under label noise conditions show that WAR outperforms the state-of-the-art in providing robust class predictions.

## 2 WASSERSTEIN ADVERSARIAL REGULARIZATION FOR LABEL NOISE

Given a set of labeled data $\{\boldsymbol{x}_i, \boldsymbol{y}_i\}_{i=1,\dots N}$, we are interested in learning a classifier $p_\theta$ defined by a set of parameters $\theta$. Data $\boldsymbol{x}_i$ are usually elements of $\mathbb{R}^d$, while $\boldsymbol{y}_i \in \mathcal{C}$ are one-hot vectors encoding the belonging to one of $C$ classes. The empirical risk minimization principle is used to learn $p_\theta$. Given a loss function $L$, the optimal set of parameters for the classifier is given by $\arg\min_\theta \sum_{i=1}^N L(\boldsymbol{x}_i, \boldsymbol{y}_i, p_\theta)$. Without loss of generality, we will consider that $L$ is the cross-entropy loss: $L_{\text{CE}}(\boldsymbol{x}_i, \boldsymbol{y}_i, p_\theta) = -\sum_c \boldsymbol{y}_i^{(c)} \log p_\theta(\boldsymbol{x}_i)^{(c)}$.

The considered label noise problem arises whenever some elements $\boldsymbol{y}_i$ do not match the real class of $\boldsymbol{x}_i$. Several scenarii exist: in the *symmetric* label noise, labels can be flipped uniformly across all the classes, whereas in the *asymmetric* label noise, labels $\boldsymbol{y}$ in the training set can be flipped with higher probability toward specific classes. We note that the first scenario, while thoroughly studied in the literature, is highly improbable in real situations: for example, it is more likely that an annotator mislabels two breeds of dogs than a dog and a car. Hence, noise in labels provided by human annotators is not symmetric since annotators make mistakes depending on class similarities (Misra et al., 2016).

To prevent a classifier to overfit on noisy labels, we would like to regularize its decision function in areas where the local uniformity of training labels is broken. To achieve such desired local uniformity, robust optimization can be used. This amounts to enforce that predicted labels are uniform in a local neighborhood $\mathcal{U}_i$ of data point $\boldsymbol{x}_i$. This changes the total loss function in the following way:

$$\arg\min_\theta \sum_{i=1}^N \max_{\boldsymbol{x}_i^u \in \mathcal{U}_i} L(\boldsymbol{x}_i, \boldsymbol{y}_i, p_\theta) \tag{1}$$

### 2.1 ADVERSARIAL REGULARIZATION

Because the robust optimization problem in equation 1 is hard to solve exactly, adversarial training (Goodfellow et al., 2015; Shaham et al., 2015) was proposed as a possible surrogate. Instead of solving the `max` inner problem, it suggests to replace it by enforcing uniformity in the direction of maximum perturbation, called the adversarial direction. Mainly used for robustness against adversarial examples, adversarial training is however not adapted to our problem, since it can enforce uniformity around a false label. Following the same reasoning, we propose to cast the problem as a regularization term of the initial loss function. The total learning loss is then

$L_{tot}(\boldsymbol{x}_i, \boldsymbol{y}_i, p_\theta) = L_{CE}(\boldsymbol{x}_i, \boldsymbol{y}_i, p_\theta) + \beta R_{AR}(\boldsymbol{x}_i, p_\theta)$, where $R_{AR}$ is a regularization term reading:

$$R_{AR}(\boldsymbol{x}_i, p_\theta) = D(p_\theta(\boldsymbol{x}_i + \mathrm{r}_i^a), p_\theta(\boldsymbol{x}_i)) \ \text{ with } \mathrm{r}_i^a = \underset{\mathrm{r}_i, \|\mathrm{r}_i\| \leq \varepsilon}{\operatorname{argmax}} D(p_\theta(\boldsymbol{x}_i + \mathrm{r}_i), p_\theta(\boldsymbol{x}_i)). \quad (2)$$

Basically, it minimizes an isotropic divergence $D$ between the probability output $p_\theta(\boldsymbol{x}_i + \mathrm{r}^a)$ and $p_\theta(\boldsymbol{x}_i)$. A sound choice for $D$ can be the Kullback-Leibler (KL) divergence. $R_{AR}$ can be seen as a (negative) measure of local smoothness , or also as a Local Lipschitz constant in the $\varepsilon$ neighborhood of $\boldsymbol{x}_i$ with respect to the metric $D$, hence a measure of complexity of the function. $R_{AR}$ promotes local uniformity in the predictions without using the potentially noisy label $\boldsymbol{y}_i$: therefore, it reduces the influence of noisy labels, since it is computed from the prediction $p_\theta(\boldsymbol{x}_i)$ that can be correct when the true label is not. $R_{AR}$ shares strong similarities with the Virtual Adversarial Training (VAT) from Miyato et al. (2018), at the notable exception that we do not consider a semi-supervised learning problem and that we regularize on the labeled training positions $\boldsymbol{x}_i$, where VAT is applied on unlabeled samples.

It can be shown (proof in the supplementary material A.1) that this regularization acts as a label smoothing technique:

**Lemma 1** *Let $D$ be the Kullback-Leibler divergence. Let $\epsilon = \frac{\beta}{\beta+1} \in [0, 1[$. Let $\boldsymbol{y}_i^a = p_\theta(\boldsymbol{x}_i + \mathrm{r}_i^a)$ be the predicted (smooth) adversarial label. Let $H$ be the entropy. The regularized learning problem $L_{tot}(\boldsymbol{x}_i, \boldsymbol{y}_i, p_\theta)$ is equivalent to :*

$$L_{tot}(\boldsymbol{x}_i, \boldsymbol{y}_i, p_\theta) \equiv L_{CE}(\boldsymbol{x}_i, (1 - \epsilon)\boldsymbol{y}_i + \epsilon \boldsymbol{y}_i^a, p_\theta) - \epsilon H(\boldsymbol{y}_i^a).$$

This leads to the following interpretation: instead of learning over the exact label or over a mix between the exact label and the network prediction, we learn over an interpolation with the adversarial label, while maximizing the entropy of the adversarial label (i.e. blurring the boundaries of the classifier). Related developments can be found in adversarial label smoothing (ALS) (Shafahi et al., 2018; Goibert & Dohmatob, 2019), which aims at providing robustness against adversarial attacks.

Yet, one of the major limit of this approach is that the regularization is conducted with the same magnitude everywhere. As a consequence, a strong regularization can remove the label noise, but also hinder the ability of the classifier to separate similar classes where a complex boundary is needed. To overcome this issue, we propose to replace $D$ by a geometry-aware divergence taking into account the specific relationships between the classes.

## 2.2 WASSERSTEIN ADVERSARIAL REGULARIZATION (WAR)

To make the divergence aware of specific relationships between classes, we replace the isotropic divergence $D$ with a Wasserstein distance computed in the labels space. We name our proposed method Wasserstein Adversarial Regularization. Frogner et al. (2015) already used the Wasserstein distance as a loss in a learning system between the output of the model for multi-label learning. The interest of the Wasserstein distance is to take into consideration the geometry of the label space.

We define the proposed regularization term $R_{WAR}$ as follows:

$$R_{WAR}(\boldsymbol{x}_i) = OT_C^\lambda(p_\theta(\boldsymbol{x}_i + \mathrm{r}_i^a), p_\theta(\boldsymbol{x}_i)) \ \text{ with } \mathrm{r}_i^a = \underset{\mathrm{r}_i, \|\mathrm{r}_i\| \leq \varepsilon}{\operatorname{argmax}} OT_C^\lambda(p_\theta(\boldsymbol{x}_i + \mathrm{r}_i), p_\theta(\boldsymbol{x}_i)). \quad (3)$$

$OT_C^\lambda$ is an optimal transport (OT) distance (Peyré & Cuturi, 2019). The OT problem seeks an optimal coupling $\boldsymbol{T}^* \in \boldsymbol{U}(\boldsymbol{\alpha}, \boldsymbol{\beta}) = \{\boldsymbol{T} | \boldsymbol{T} \geq \boldsymbol{0}, \boldsymbol{T}\boldsymbol{1} = \boldsymbol{\alpha}, \boldsymbol{T}^\top \boldsymbol{1} = \boldsymbol{\beta}\}$ between two distributions $\boldsymbol{\alpha}, \boldsymbol{\beta}$ with respect to a ground cost $\boldsymbol{C} \in \mathbb{R}^{n_1 \times n_2}$. $\boldsymbol{U}(\boldsymbol{\alpha}, \boldsymbol{\beta})$ is the space of joint probability distributions with marginals $\boldsymbol{\alpha}$ and $\boldsymbol{\beta}$. OT distances are classically expressed through the Wasserstein distance $W_C(\boldsymbol{\alpha}, \boldsymbol{\beta}) = \underset{\boldsymbol{T} \in \boldsymbol{U}(\boldsymbol{\alpha}, \boldsymbol{\beta})}{\min} \langle \boldsymbol{T}, \boldsymbol{C} \rangle$, where $\langle ., . \rangle$ is the Frobenius product. Unfortunately, this distance is expensive to compute (cubical complexity). In practice, we will use the solution of the sharp entropic variant of the optimal transport problem (Luise et al., 2018):

$$OT_C^\lambda(\boldsymbol{\alpha}, \boldsymbol{\beta}) = \langle \boldsymbol{T}_\lambda^*, \boldsymbol{C} \rangle \ \text{ with } \boldsymbol{T}_\lambda^* = \underset{\boldsymbol{T} \in \boldsymbol{U}(\boldsymbol{\alpha}, \boldsymbol{\beta})}{\operatorname{argmin}} \langle \boldsymbol{T}, \boldsymbol{C} \rangle - \lambda H(\boldsymbol{T})$$

where $H$ denotes the entropy function and $\lambda$ the regularization strength. Using this regularized version has several advantages: *i)* it lowers the computational complexity to quadratic, *ii)* it turns the

problem into a strongly convex one, for which gradients can be computed efficiently and *iii)* it allows to vectorize the computation of all Wasserstein distances in a batch, which is particularly appealing for training deep neural nets. Based on (Genevay et al., 2018), we use the *AutoDiff* framework, which approximates the derivative of this regularization with a fixed number of iterations of the Sinkhorn algorithm.

**Choice of ground cost.** The ground cost $C$ reflects the geometry of the label space. It bridges the gap between AR and WAR. An uninformative 0-1 ground cost, *i.e.* 0 over the diagonal and 1 everywhere, would give the total variation (TV) loss (Remark 2.26 in (Peyré & Cuturi, 2019)), which could also be used as $D$ in the AR framework. Below, we refer to this special case as $R_{\text{WAR.0-1}}$. To define a $C$ matrix encoding class relations, multiple choices are possible. We could calculate the Wasserstein distance between classes, but this would be biased as we have noisy labels. We could set it manually (Tuia et al., 2011), but this becomes unpractical when a large set of classes is present. In absence of prior information about the nature of the source of labelling errors, we decided to rely on semantic distances based on word embeddings such as *word2vec* (Mikolov et al., 2013). Similarities between classes are then defined via Euclidean distances in the embedding space, as proposed in (Frogner et al., 2015). Finally, as our method requires large values of the cost between similar classes, we apply the function $e^{-m}$ (where $m$ is the Euclidean distance between the two class names) element-wise and set the diagonal of $C$ to 0.

**Function smoothness and ground metric.** Now we discuss how the proposed regularization term regularizes the model $p_\theta$ with a smoothness controlled by the ground metric $C$. It is not possible to extend the label smoothing in Lemma 1 to WAR because the OT distance does not admit a close form solution, but we can still show how $R_{\text{WAR}}$ promotes label smoothness. To this end, we look at the regularization term $OT_C^\lambda(\hat{p}_\theta(\boldsymbol{x}), p_\theta(\boldsymbol{x} + \text{r}))$ for a given sample $\boldsymbol{x}$ and a pre-computed r. We can prove (see A.2 of the supplementary material) the following lemma:

**Lemma 2** *Minimizing $R_{WAR}$ with a symmetric cost $C$ such that $C_{i,i} = 0, \forall i$ is equivalent to minimizing a weighted total variation (TV) norm between $p_\theta(\boldsymbol{x})$ and $p_\theta(\boldsymbol{x} + \text{r})$.*

$$\underline{c}TV(p_\theta(\boldsymbol{x}), p_\theta(\boldsymbol{x} + \text{r})) \leq \sum_k \underline{c}_k |p_\theta(\boldsymbol{x})_k - p_\theta(\boldsymbol{x} + \text{r})_k| \leq OT_C^\lambda(p_\theta(\boldsymbol{x}), p_\theta(\boldsymbol{x} + \text{r})) \qquad (4)$$

*where $\underline{c}_k = \min_{i, i \neq k} c_{k,i}$ is the minimal off-diagonal cost for row $k$ of $C$ and $\underline{c} = \min_k \underline{c}_k$ is a global minimum out of the diagonal.*

By minimizing the proposed $R_{\text{WAR}}$ regularization with r belonging in a small ball around $\boldsymbol{x}$, we actually minimize a local approximation of the Lipschitz constant of $p_\theta$. This has the effect of smoothing-out the model around $\boldsymbol{x}$ and makes it more robust to label noise. One can see the effect of the cost matrix in the center term of equation 4, where the values in the ground metric correspond to a weighting of a total variation, hence controlling the effect of the regularization. Interestingly, the Wasserstein distance can be bounded both below ($\underline{c}$) and above ($\bar{c}$) by Total Variation and weighted total variation similarly to the equation above. Finally, in practice we minimize the expectation of the OT loss, which means that we will penalize areas of high density similarly to a regularization with the Sobolev norm (i.e. penalizing the expected norm of the model gradient (Mroueh et al., 2018)), while keeping a finer control of the class relations, since we use the ground loss $C$ that promotes anisotropy.

**Illustration of the effect of $R_{\text{WAR}}$** We illustrate AR and WAR in a simple toy 3 classes classification problem with noise (Figure 1).

Each column of the figure corresponds to a divergence function $D$. The top row illustrates the values on the simplex, while the bottom row shows the classification predictions when using $D$ as adversarial regularization. From left to right, we compare the effect of training with the cross entropy alone (CCE, no regularization), $R_{\text{WAR}}$ with $\lambda = 0.1$ and $\lambda = 0.05$, as well as $R_{\text{AR}}$ with TV, KL and L2 divergences as $D$. For the classification problem, we generated two close classes (in orange and red), as well as a third (in black), which is far from the others. Then, we introduced noisy labels (of the black class) in the region of the red class.

On this toy example, CCE overfits the noisy black labels, yet is able to distinguish the red and orange classes. The $R_{\text{AR}}$ regularizers, being class agnostic, correct for the noisy black labels in the bottom

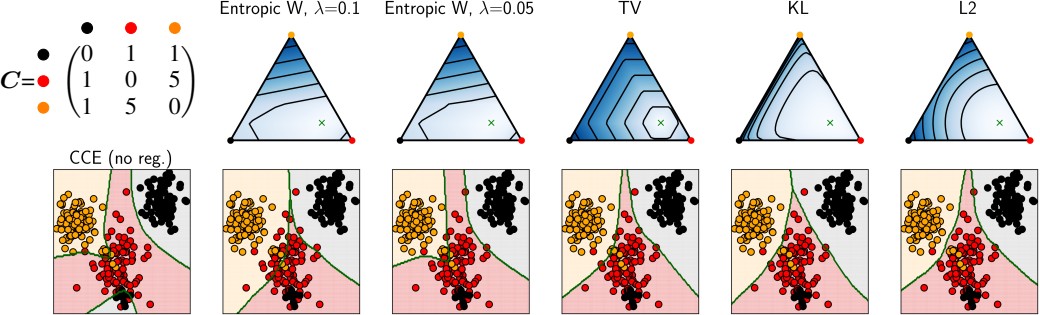

Figure 1: Illustration of the regularization geometry for different losses in the adversarial training. (Top) Regularization values on the simplex of class probabilities. Each corner stands for a class. All losses are computed with respect to a prediction represented as the green x. Colors are as follows: white is zero while darker is bigger. In the case of WAR, the ground cost $C$ is given on the left. (Down) Classification boundaries when using these losses for regularization. The unregularized classifier (CCE) is given on the left.

part, but smooth the complex decision function between the orange and red classes. On the contrary, $R_{\text{WAR}}$ uses a different cost per pair of classes, illustrated in the top left panel of Figure 1: the smallest cost is set between the red and black classes, which has the effect of promoting adversarial examples in that direction. This cost / smoothing relation is due to the fact that our problem is a minimization of the OT loss: in other words, the higher the cost between the classes, the less the binary decision boundary will be smoothed. Finally, the effect of the global $\lambda$ parameter can also be appreciated in the classification results: while using $\lambda = 0.05$, the smoothing of the loss is decreased and the final decision boundary between the mixed classes keeps all its complexity.

## 3 EXPERIMENTS

We evaluate the proposed approach WAR on both image classification and semantic segmentation tasks. We first showcase the performance of WAR on a series of image classification benchmarks (Section 3.1), and then consider two real world cases: first, the classification of clothing images from online shopping websites (Section 3.2) and then the semantic segmentation of land use in sub-decimeter resolution aerial images (Section 3.3).

### 3.1 IMAGE CLASSIFICATION ON SIMULATED BENCHMARK DATASETS

**Datasets and noisy labels simulations.** We consider three image classification benchmark datasets: Fashion-MNIST (Xiao et al., 2017), and CIFAR-10 / CIFAR-100 (Krizhevsky, 2009). Fashion-MNIST consists of $60'000$ gray scale images of size $28 \times 28$ with 10 classes. CIFAR-10 and CIFAR-100 consist of $50'000$ color images of size $32 \times 32$ covering 10 and 100 classes, respectively. Each dataset also contains $10'000$ test images with balanced classes.

Since we want to evaluate robustness to noisy labels, we simulated label noise in the training data only. For all datasets, we introduced 0%, 20% and 40% of noise in the labels. We considered only asymmetric noise, a class-conditional label noise where each label $y_i$ in the training set is flipped into $y_j$ with probability $P_{i,j}$. As described above, asymmetric noise is more common in real world scenarios than symmetric noise, where the labels are flipped uniformly over all the classes. For CIFAR-10 and CIFAR-100, we follow the asymmetric noise simulation setting by Patrini et al. (2017), where class labels are swapped only among similar classes with probability $p$ (i.e. the noise level). For Fashion-MNIST, we visually inspected the similarity between classes on a $t$-SNE plot of the activations of the model trained on clean data; we then swapped labels between overlapping classes ($\rightarrow$: one-directional swap, $\leftrightarrow$ mutual swap): DRESS $\rightarrow$ T-SHIRT/TOP, COAT $\leftrightarrow$ SHIRT, SANDAL $\rightarrow$ SNEAKER, SHIRT $\rightarrow$ PULLOVER, ANKLE BOOT $\rightarrow$ SNEAKER.

**Baselines.** We compared the proposed WAR with an informative $C$ matrix based on the `word2vec` embedding (WAR$_C$) against several state-of-the-art methods: `Unhinged` (Rooyen

Table 1: Test accuracy (%) of different models on Fashion-MNIST, CIFAR-10, and CIFAR-100 dataset with varying noise rates ($0\% - 40\%$). The mean accuracies and standard deviations averaged over the last 10 epochs of three runs are reported, and the best results are highlighted in **bold**.

| Dataset / | noise | CCE | Backward | Forward | Unhinge | Bootsoft | CoTeaching | CoTeaching+ | D2L | $\text{WAR}_C$ |
|---|---|---|---|---|---|---|---|---|---|---|
| Fashion-MNIST | 0% | 94.69±0.11 | 94.86±0.04 | 94.81±0.04 | **95.12 ± 0.03** | 94.79±0.02 | 94.28±0.04 | 93.62±0.01 | 94.47±0.02 | 94.70±0.02 |
| | 20% | 89.02±0.47 | 88.84±0.10 | 91.03±0.12 | 90.04± 0.08 | 88.17±0.11 | 91.24±0.06 | 92.26±0.02 | 89.12±0.15 | **93.37±0.08** |
| | 40% | 78.85±0.56 | 81.74±0.08 | 82.85 ±0.2 | 78.32 ±0.15 | 73.84±0.28 | 86.83±0.10 | 86.15±0.03 | 78.98±0.25 | **90.41±0.02** |
| CIFAR-10 | 0% | 91.76±0.04 | 91.63 ± 0.04 | 91.59 ± 0.03 | **92.27 ± 0.04** | 91.67 ± 0.03 | 90.12 ±0.04 | 88.47±0.14 | 91.29±0.02 | 91.88±0.31 |
| | 20% | 85.26±0.09 | 84.67 ± 0.1 | 85.70 ± 0.08 | 87.09 ± 0.05 | 85.35 ± 0.8 | 86.19 ±0.07 | 82.97±0.25 | 86.64 ±0.12 | **89.12±0.48** |
| | 40% | 76.23±0.15 | 73.49 ± 0.14 | 75.10 ± 0.15 | 77.94 ± 0.1 | 74.32 ± 0.2 | 80.87±0.09 | 72.65±0.10 | 73.12±0.43 | **84.55±0.78** |
| CIFAR-100 | 0% | 68.60±0.09 | 69.53±0.07 | 70.12±0.07 | 70.54±0.06 | 69.81±0.04 | 65.42±0.06 | 58. 93±0.14 | **70.93 ±0.02** | 68.16±0.18 |
| | 20% | 58.81±0.10 | 59.23±0.08 | 59.54±0.05 | 61.06±0.06 | 58.97±0.08 | 56.55±0.08 | 44.88±0.14 | 60.90±0.03 | **62.72±0.16** |
| | 40% | 42.45±0.12 | 43.02±0.09 | 42.17±0.1 | 42.87±0.07 | 41.73±0.08 | 42.73±0.08 | 29.94±0.34 | 42.61±0.04 | **58.86±0.21** |
| Avg,. rank | | 5.9 | 5.2 | 4.6 | 3.0 | 6.2 | 5.2 | 7.6 | 5.1 | **2.2** |
| Avg,. rank (noise only) | | 6.3 | 5.8 | 4.8 | 3.8 | 7.3 | 3.8 | 6.8 | 5.2 | **1.0** |

et al., 2015), `Bootstrapping` (Reed et al., 2015), `Forward` and `Backward` loss correction (Patrini et al., 2017), `dimensionality driven learning (D2L)` (Ma et al., 2018), `Co-Teaching` (Han et al., 2018), and `Co-Teaching+` (Yu et al., 2019). Finally, as a baseline for all the considered methods we also included a `categorical cross entropy (CCE)` loss function. All the methods shared the same architecture and training procedures, as detailed in the supplementary material A.4.

**Model.** Similarly to other works (Han et al., 2018; Miyato et al., 2018), we employed a 9-layer CNN architecture, detailed in Table 5 of the supplementary material A.4. For $\text{WAR}_C$, we set the hyper-parameters $\beta= 10$, $\lambda= 0.05$, and $\varepsilon= 0.005$ for all the datasets. The hyper-parameters of the baselines are set according to their original papers. The noise transition matrix for the `Forward` and `Backward` method is estimated from the model trained with cross entropy (Patrini et al., 2017). The source code of WAR in PyTorch (Paszke et al., 2017) will be released upon publication.

**Results.** Classification accuracies are reported in Table 1. Results show that $\text{WAR}_C$ consistently outperforms the competitors by large margins, across noise levels and datasets. In particular, $\text{WAR}_C$ achieved improvements of 4-5% points on fashion-MNIST/CIFAR-10, and 15% on CIFAR-100 at the highest noise level. This demonstrates that the inclusion of class geometric information during training mitigates the effect of over-fitting to noisy labels. Besides $\text{WAR}_C$, `Unhinged` and `Co-Teaching` also performed well. The `Forward` and `Backward` method performed slightly better than `CCE`, which is most likely due to the burden in accurately estimating the noise transition matrix. It is noted that `Co-Teaching` uses true noise estimate, and the accuracy might drop if the noise ratio is estimated directly from the noisy data. Furthermore, performance of `Co-Teaching+`[1] is surprising lower than the one of `Co-Teaching` on two datasets, in contrast to the observations in (Yu et al., 2019). From our experiments, we observed that `Co-Teaching+` does not perform well when the noise is class-dependent and the model considered has a wide capacity. We provide a detailed discussion about `Co-Teaching+` in the supplementary material (section A.7).

**Importance of encoding class similarities:** Finally, to better assess the significance of including class similarities, we compared $\text{WAR}_C$ with $\text{WAR}_{0-1}$ and `AR` using KL divergence (Miyato et al., 2018). For `AR`, we did not use the hyper-parameters from the original paper, as it would lead to over-fitting on the noisy labels in the later stage of training. Therefore, we used $\beta=5$ and $\varepsilon = 0.005$ (similarly to both `WAR` approaches), and followed the same training procedure as `WAR` (as detailed in A.4, where a sensitivity analysis for $\beta$ is also reported in A.5).

Table 2 reports the performance of $\text{WAR}_{0-1}$ and `AR`, and shows that $\text{WAR}_C$ is consistently better than `AR` and $\text{WAR}_{0-1}$ (except in one case), and outperformed `AR` significantly by a 2-3% margin at the highest noise level. This demonstrates that including priors about class similarities as done in $\text{WAR}_C$ helps increasing the robustness against label noise.

---

[1]We used the code provided by the authors: https://github.com/xingruiyu/coteaching_plus

Table 2: Comparison of $\text{WAR}_C$ with AR and $\text{WAR}_{0-1}$ with varying noise rates ($0\% - 40\%$). The mean accuracies and standard deviations averaged over the last 10 epochs of three runs are reported, and the best results are highlighted in **bold**.

| Methods | Fashion-MNIST | | | CIFAR-10 | | | CIFAR-100 | | |
|---|---|---|---|---|---|---|---|---|---|
| | 0% | 20% | 40% | 0% | 20% | 40% | 0% | 20% | 40% |
| AR | **94.81±0.09** | 93.10±0.14 | 89.74±0.10 | 91.49±0.07 | 88.91±0.09 | 81.98±0.25 | 67.83±0.10 | **65.44±0.11** | 55.75±0.14 |
| $\text{WAR}_{0-1}$ | 94.60±0.03 | 90.99±0.07 | 86.03±0.20 | 90.94±0.12 | 86.12±0.21 | 74.15±0.34 | 65.78±0.15 | 60.56±0.14 | 51.00±0.31 |
| $\text{WAR}_C$ | 94.70±0.02 | **93.37±0.08** | **90.41±0.02** | **91.88±0.31** | **89.12±0.48** | **84.55±0.78** | **68.16±0.18** | 62.72±0.16 | **58.86±0.21** |

Table 3: Test accuracy of different models on Clothing1M dataset with ResNet-50.

| Methods | CCE | CCE (reproduced) | bootsoft | Forward | D2L | $\text{WAR}_C$ |
|---|---|---|---|---|---|---|
| accuracy | 68.80 | 68.63 | 68.94 | 69.84 | 69.47 | **70.66** |

## 3.2 IMAGE CLASSIFICATION WITH REAL-WORLD LABEL NOISE

**Dataset.**   In this section, we demonstrate the robustness of $\text{WAR}_C$ on a large scale real world noisy label dataset, Clothing1M (Xiao et al., 2015). The Clothing1M dataset contains 1 million images of clothing obtained from online shopping websites and has 14 classes. The labels have been obtained from text surrounding the images and are thus extremely noisy. The overall accuracy of the labels has been estimated to $\approx 61.54\%$. The dataset also contains additional manually refined clean data for training (50k samples), validation (14k) and testing (10k). However, we did not use the clean training and validation data in this work. Only the testing clean data was used to evaluate the performance of the different approach when learning with label noise.

**Experimental setup and Results:**   Similar to (Patrini et al., 2017; Wang et al., 2019), we used ResNet-50 pre-trained on ImageNet for a fair comparison between methods (more details in A.4). For $\text{WAR}_C$, the hyperparameters are similar to those of the previous experiment, except $\varepsilon = 0.05$. Results are reported in Table 3, where we compared $\text{WAR}_C$ against the competitors from (Patrini et al., 2017; Wang et al., 2019), and also reproduced the CCE accuracy by our own experiments. As shown in Table 3, our method achieved the highest performance compared to all the baselines.

## 3.3 SEMANTIC SEGMENTATION OF AERIAL IMAGES

**Datasets and noisy labels simulations.**   In this experiment we consider the task of assigning every pixel of an aerial image to an urban land use category. We considered a widely used remote sensing benchmark, the ISPRS Vaihingen semantic labeling dataset[2]. The data consist of 33 tiles (of varying sizes, for a total of $168'287'871$ pixels) acquired by an aircraft at the ground resolution of 9cm. The images are true orthophotos with three spectral channels (near infrared, red, green). A digital surface model (DSM) and a normalized digital surface model (nDSM) are also available, making the input space 5-dimensional. Among the 33 tiles, we used the initial data split (11 tiles for training, 5 for validation and 17 for testing). As ground truth, six land cover classes (impervious surfaces, building, low vegetation, tree, car, background/clutter) are densely annotated.

We simulated label noise by swapping labels at the object level rather than flipping single pixels. An object is the connected component of pixels sharing the same label. We also focused on plausible labeling errors: for instance, a car could be mislabeled to an impervious surface, but not to a building or a tree. Following this methodology, a third of the connected components had the label flipped. An example of the corrupted data is shown in Figure 3 in the supplementary material (A.6).

**Model.**   We used a U-Net architecture (Ronneberger et al., 2015), modified to take the 5 channels input data as inputs. More details about the training procedure are reported in the supplementary material A.4. Using this methodology, we obtain an overall accuracy on the clean data of 83.89%, which is close to the state of the art for this dataset.

---

[2]http://www2.isprs.org/commissions/comm3/wg4/2d-sem-label-vaihingen.html

Table 4: Per class F1 scores, average F1 score and overall accuracy (%) on the test set of Vaihingen. The best results (on the noisy dataset) are highlighted in **bold**.

| Class | CCE | CCE | Bootsoft | CoTeaching | AR | WAR |
|---|---|---|---|---|---|---|
| Training set | Clean | | | Noisy | | |
| Buildings | 90.29 | 75.06 | 88.34 | 75.1 | 81.6 | **89.04** |
| Cars | 58.91 | 14.21 | 10.98 | **26.6** | 21.6 | 25.78 |
| Impervious surfaces | 85.76 | 62.20 | **82.66** | 76.9 | 70.9 | 79.01 |
| Low vegetation | 76.32 | 25.92 | 61.40 | 57.4 | 57.8 | **71.56** |
| Trees | 84.72 | 70.89 | 80.49 | 77.5 | 78.9 | **82.92** |
| Average F1 | 79.20 | 49.65 | 64.77 | 63.6 | 62.2 | **69.66** |
| Overall accuracy | 83.89 | 63.95 | **78.87** | 74.6 | 77.1 | 78.43 |

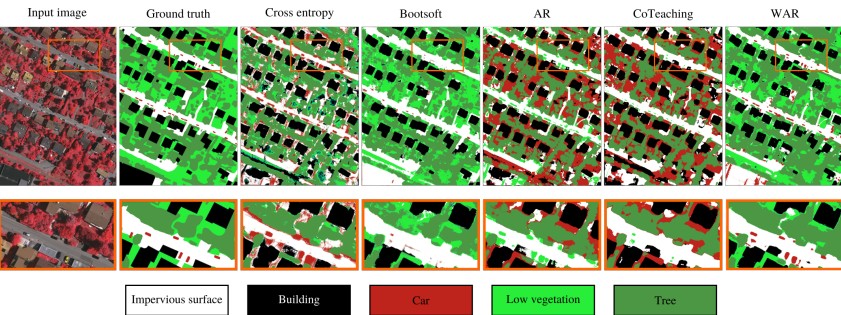

Figure 2: Semantic segmentation maps obtained on the test set of the ISPRS Vaihingen dataset (tile #12 of the original data). The top row shows the full image, and the second row shows a close-up of the area delineated in orange.

**Results.** We compare $\text{WAR}_C$ with standard CCE, Bootsoft, Co-Teaching and AR. The results, computed on the full test ground truth (including boundaries) and averaged over 2 runs, are reported in Table 4. Note that the classes are unbalanced and, for most of them, the F1-score is improved using $\text{WAR}_C$, except for the dominant class (impervious surfaces). This leads to a much higher average F1-score using $\text{WAR}_C$ (compared to its competitors), while the overall accuracy is only slightly decreased compared to Bootsoft. This behavior can be seen in the maps shown in Figure 2. We can see in the close-ups that Bootsoft performs poorly in detecting the cars, which are often confused with generic impervious surfaces.

## 4 CONCLUSION AND DISCUSSION

In this paper, we proposed Wasserstein Adversarial Regularization (WAR) to address the problem of learning with noisy labels. Using a ground cost based on class similarites or prior knowledge, we are able to change the geometry of the regularization loss according to class similarities. We compare WAR with state of the art algorithms on the Fashion-MNIST, CIFAR-10, CIFAR-100 benchmarks with noisy labels up to 40%. WAR outperformed state of the art results on all benchmarks. Furthermore, we proved that WAR performs accurately on real life problems in both classification and semantic segmentation problems.

Future works will consider exploring other strategies to define the ground cost, beyond the current 'a priori' setting: such cost could be for instance learned from the data structure itself. Moreover, even if in this paper we focused on the label noise problem, WAR remains a generic regularization scheme that could be applied to other classical learning problems as enforcing adversarial robustness or semi-supervised learning.

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

## A  SUPPLEMENTARY MATERIAL

### A.1  LINKS BETWEEN AR AND ADVERSARIAL LABEL SMOOTHING

This part gathers the proof for Lemma 1. The total learning loss for one sample $(\boldsymbol{x}, \boldsymbol{y})$ is:

$$L_{\text{tot}}(\boldsymbol{x}, \boldsymbol{y}, p_\theta) = L_{\text{CE}}(\boldsymbol{x}, \boldsymbol{y}, p_\theta) + \beta R_{\text{AR}}(\boldsymbol{x}, p_\theta) \tag{5}$$

where $L_{\text{CE}}$ is the cross entropy loss:

$$L_{\text{CE}}(\boldsymbol{x}, \boldsymbol{y}, p_\theta) = -\sum_c \boldsymbol{y}^{(c)} \log p_\theta(\boldsymbol{x})^{(c)} \tag{6}$$

We write the $R_{\text{AR}}$ regularization as:

$$R_{\text{AR}}(\boldsymbol{x}, p_\theta) = D_{KL}(p_\theta(\boldsymbol{x} + \text{r}^a), p_\theta(\boldsymbol{x}))$$
$$\text{where } \text{r}^a = \underset{\text{r}, \|\text{r}\| \leq \varepsilon}{\operatorname{argmax}} D_{KL}(p_\theta(\boldsymbol{x} + \text{r}), p_\theta(\boldsymbol{x})). \tag{7}$$

We have that:

$$D_{KL}(p_\theta(\boldsymbol{x} + \text{r}^a), p_\theta(\boldsymbol{x})) = \sum_c p_\theta(\boldsymbol{x} + \text{r}^a)^{(c)} \log \frac{p_\theta(\boldsymbol{x} + \text{r}^a)^{(c)}}{p_\theta(\boldsymbol{x})^{(c)}}$$

$$= \sum_c p_\theta(\boldsymbol{x} + \text{r}^a)^{(c)} \log p_\theta(\boldsymbol{x} + \text{r}^a)^{(c)} - \sum_c p_\theta(\boldsymbol{x} + \text{r}^a)^{(c)} \log p_\theta(\boldsymbol{x})^{(c)}$$

$$= -\sum_c p_\theta(\boldsymbol{x} + \text{r}^a)^{(c)} \log p_\theta(\boldsymbol{x})^{(c)} - H(p_\theta(\boldsymbol{x} + \text{r}^a)). \tag{8}$$

where $H$ is the entropy function. Consequently, the total loss can be rewritten as:

$$L_{\text{tot}}(\boldsymbol{x}, \boldsymbol{y}, p_\theta) = -\sum_c (\boldsymbol{y}^{(c)} + \beta p_\theta(\boldsymbol{x} + \text{r}^a)^{(c)}) \log p_\theta(\boldsymbol{x})^{(c)} - \beta H(p_\theta(\boldsymbol{x} + \text{r}^a)) \tag{9}$$

Here $\beta \in \mathbb{R}^+$. Let $\beta = \frac{\epsilon}{1-\epsilon}$ with $\epsilon \in [0, 1[$. We have the following equivalence:

$$(1 - \epsilon) L_{\text{tot}}(\boldsymbol{x}, \boldsymbol{y}, p_\theta) = -\sum_c ((1 - \epsilon)\boldsymbol{y}^{(c)} + \epsilon p_\theta(\boldsymbol{x} + \text{r}^a)^{(c)}) \log p_\theta(\boldsymbol{x})^{(c)} - \epsilon H(p_\theta(\boldsymbol{x} + \text{r}^a))$$

$$= L_{\text{CE}}(\boldsymbol{x}, \underbrace{(1 - \epsilon)\boldsymbol{y} + \epsilon p_\theta(\boldsymbol{x} + \text{r}^a))}_{\text{Interpolated label}}, p_\theta) - \epsilon \underbrace{H(p_\theta(\boldsymbol{x} + \text{r}^a))}_{\text{adversarial label entropy}} \tag{10}$$

*i.e.* That's why learning with the total loss is equivalent to learn on an interpolated label and maximizing the entropy of adversarial labels.

### A.2  LINKS BETWEEN WAR AND LABEL SMOOTHING

In this subsection we aim to prove the following relations

$$\underline{c}TV(\boldsymbol{\alpha}, \boldsymbol{\beta})) \leq \sum_i \underline{c}_i |\alpha_i - \beta_i| \leq OT_{\boldsymbol{C}}^\lambda(\boldsymbol{\alpha}, \boldsymbol{\beta})$$

First we recall the definition of the Wasserstein distance

$$W_{\boldsymbol{C}}(\boldsymbol{\alpha}, \boldsymbol{\beta}) = \min_{\boldsymbol{T} \in \boldsymbol{U}(\boldsymbol{\alpha}, \boldsymbol{\beta})} \langle \boldsymbol{T}, \boldsymbol{C} \rangle \tag{11}$$

where $\boldsymbol{U}(\boldsymbol{\alpha}, \boldsymbol{\beta}) = \{\boldsymbol{T} | \boldsymbol{T} \geq \boldsymbol{0}, \boldsymbol{T}\boldsymbol{1} = \boldsymbol{\alpha}, \boldsymbol{T}^\top \boldsymbol{1} = \boldsymbol{\beta}\}$. It is well known and obvious from (Cuturi, 2013; Luise et al., 2018) that the optimal OT matrix of regularized OT $\boldsymbol{T}_\lambda^\star$ (Equation 4) leads to a larger OT loss than the exact OT solution of the problem above $\boldsymbol{T}^\star$. this means that

$$W_{\boldsymbol{C}}(\boldsymbol{\alpha}, \boldsymbol{\beta}) = \langle \boldsymbol{T}^\star, \boldsymbol{C} \rangle \leq \langle \boldsymbol{T}_\lambda^\star, \boldsymbol{C} \rangle = OT_{\boldsymbol{C}}^\lambda(\boldsymbol{\alpha}, \boldsymbol{\beta}) \tag{12}$$

and the relation is strict when $\lambda > 0$.

Now if we suppose that the cost matrix is symmetric and $C_{i,i} = 0$ and $C_{i,j} > 0$ when $i \neq j$ then solving equation 11 means that the maximum amount of mass on the diagonal of $T^\star$ since it leads to a 0 cost. Under constraints $U(\boldsymbol{\alpha}, \boldsymbol{\beta})$ this maximum amount is equal to $T^\star_{i,i} = \min(\alpha_i, \beta_i), \forall i$. This implies that for a given row $i$ in $T^\star$ the amount of mass not on the diagonal is for row $i$ is $\sum_{j \neq i} T^\star_{i,j} = \max(\alpha_i - \beta_i, 0)$ because of the left marginal constraint in $U$. Note that a similar result can be expressed with the column j such that the mass not on the diagonal of column $j$ is $\sum_{i \neq j} T^\star_{i,j} = \max(\beta_j - \alpha_j, 0)$. This obviously means that for a given column/row index $k$ we have $\sum_{i \neq k} T^\star_{i,k} + T^\star_{k,i} = |\alpha_k - \beta_k|$.

Let's write $A_k = \sum_{i \geq k, j \geq k} T^\star_{i,j} C_{i,j}$. We have that $W_C(\boldsymbol{\alpha}, \boldsymbol{\beta}) = A_1$. Now we remark that

$$A_k - A_{k+1} \geq \underline{c}_k |\alpha_k - \beta_k|.$$

We can write that

$$A_1 = A_1 - \sum_{k=2} A_k + \sum_{k=2} A_k.$$

Since $A_N = 0$ because $C_{N,N} = 0$, it turns out that $A_1 = \sum_{k=1}(A_k - A_{k+1})$. Lower bounding every elements of the sum by the previous minoration gives that:

$$A_1 = \sum_{k=1}(A_k - A_{k+1}) \tag{13}$$

$$\geq \sum_{k=1} \underline{c}_k |\alpha_k - \beta_k| \tag{14}$$

$$\geq \underline{c} TV(\boldsymbol{\alpha}, \boldsymbol{\beta}) \tag{15}$$

which gives the desired results.

## A.3 ADVERSARIAL SAMPLES COMPUTATION FOR WAR

WAR requires an efficient computation of adversarial samples. Following Miyato et al. (2018), we choose the following computation model. One could use the gradient with respect to the input r but because of differentiability, it vanishes in r = 0. When we approximate $OT^\lambda_C$ in r = 0 through the second order Taylor expansion, we have

$$OT^\lambda_C(p_\theta(\boldsymbol{x}), p_\theta(\boldsymbol{x} + \mathrm{r})) \underset{r=0}{\sim} \frac{1}{2} \mathrm{r}^t \boldsymbol{H}_\mathrm{r} \mathrm{r}. \tag{16}$$

However, computing the hessian $\boldsymbol{H}_\mathrm{r}$ with respect to r = 0 is costly. Instead we use the power iteration method (Golub & van der Vorst, 2000) to estimate the dominant hessian's eigenvector that represent the direction in which the classification function will change the most. The algorithm is repeated $k_{\max}$ times, but both the literature and our results suggest that only one iteration is sufficient to achieve state of the art results. Once the adversarial direction $\boldsymbol{d}$ is defined, one can obtain the adversarial example with r $= \varepsilon \boldsymbol{d} / \|\boldsymbol{d}\|_2$, by projecting onto the ball of radius $\varepsilon$.

## A.4 MODEL ARCHITECTURE, IMPLEMENTATION DETAILS AND TRAINING PROCEDURE

### A.4.1 BENCHMARK DATASETS

We have used a 9 layer CNN following Han et al. (2018) for the three image classification benchmark datasets: Fashion-MNIST, CIFAR10, and CIFAR100 as shown in Table 5. Between each layer we use a batch norm layer, a drop-out layer and a leaky-relu activation function with slope of $0.01$. We use the Adam optimizer for all our networks with an initial learning rate of 0.001 with coefficient $(\beta_1, \beta_2) = (0.9, 0.999)$ and with mini-batch size of 256. The learning rate is divided by 10 after epochs 20 and 40 for Fashion-MNIST (60 epochs in total), after epochs 40 and 80 for CIFAR-10 (120 epochs in total), and after epochs 80 and 120 for CIFAR-100 (150 epochs in total). While training WAR, we set $\beta = 0$ for 15 epochs for faster convergence, as we observed that the network does not overfit on noisy labels at early stages of training. The input images are scaled between [-1, 1] for Fashion-MNIST, and mean subtracted for the CIFAR10, and CIFAR100 datasets before feeding into the network. The proposed method WAR, AR, and cross entropy loss functions are

implemented in PyTorch, and for the Co-teaching[3], Co-teaching+[4] method we used the PyTorch code provided by the authors. For the rest of the state-of-the-art methods (dimensionality driven learning[5], forward and backward loss correction[6], and robust loss functions[7]: unhinged and boot strapping) the experiments are conducted using the Keras code provided by respective authors. We used similar layer initialization for all the methods in Pytorch and Keras.

| Fashion-MNIST | CIFAR-10 | CIFAR-100 |
|---|---|---|
| $28\times28\times1$ | $32\times32\times3$ | $32\times32\times3$ |
| 3×3 conv, 128 LReLU | | |
| 3×3 conv, 128 LReLU | | |
| 3×3 conv, 128 LReLU | | |
| 2×2 max-pool, stride 2 | | |
| dropout, p=0.25 | | |
| 3×3 conv, 256 LReLU | | |
| 3×3 conv, 256 LReLU | | |
| 3×3 conv, 256 LReLU | | |
| 2×2 max-pool, stride 2 | | |
| dropout, p=0.25 | | |
| 3×3 conv, 512 LReLU | | |
| 3×3 conv, 256 LReLU | | |
| 3×3 conv, 128 LReLU | | |
| avg-pool | | |
| dense $128 \to 10$ | dense $128 \to 10$ | dense $128 \to 100$ |

Table 5: CNN models used in our experiments on Fashion-MNIST, CIFAR-10 and CIFAR-100.

### A.4.2 CLOTHING 1M DATASET

Regarding the training procedure for the Clothing1M dataset we give the following details. Data pre-processing includes resizing the image to 256 x 256, center cropping a 224 x 224 patch from the resized image, and performing mean subtraction. We used a batch size of 32 and learning rate of 0.0008 to update the network with SGD optimizer with a momentum of 0.9. The learning rate is divided by 10 after 3 epochs (5 epochs in total).

### A.4.3 VAIHINGEN DATASET

We now give the training procedure for our U-net on the ISPRS Vaihingen semantic labeling dataset. The network was trained for 300 epochs (with $90°$, $180°$ or $270°$ rotations and vertical or horizontal flips as data augmentation) using the Adam optimizer with an initial learning rate of $10^{-4}$ and coefficients $(\beta_1, \beta_2) = (0.9, 0.999)$. After 10 epochs, the learning rate is set to $10^{-5}$. Furthermore, we predict on the full image using overlapping patches (200 pixels overlap) averaged according to a Gaussian kernel centered in the middle of the patch ($\sigma = 1$).

### A.5 SENSITIVITY ANALYSIS OF $\beta$ IN WAR, WAR$_{0-1}$, AND AR

We conducted an experimental study to analysis the sensitivity of the trade-off parameter ($\beta$) between the cross entropy and the adversarial regularization term on CIFAR-10 dataset with 40% noise level. The experimental results with different values of $\beta$ are shown in Table 6 and the result reveals that as $\beta$ increases, AR and WAR are robust to the label noise. However for the higher $\beta$, AR does over-smoothing and decreases the classification accuracy. On the other hand, WAR increases the accuracy as $\beta$ increases. This behaviour shows the capability of our proposed method WAR to preserve the discrimination capability between similar classes. It is noted that WAR points the

---

[3]https://github.com/bhanML/Co-teaching

[4]https://github.com/xingruiyu/coteaching_plus

[5]https://github.com/xingjunm/dimensionality-driven-learning

[6]https://github.com/giorgiop/loss-correction

[7]https://github.com/giorgiop/loss-correction

gradient direction towards the low cost classes, as a result it does not over smooths between the conflicting classes, thus maintaining the discrimination ability. Furthermore, $WAR_{0-1}$ with uninformative ground cost did not provide better results, and it is mostly similar with different values of $\beta$. This observation reinstates need of having meaningful ground cost to capture the relationship between the classes in the dataset, and to guide gradient direction with respect to the ground cost.

| $\beta$ | AR | $WAR_{0-1}$ | WAR |
|---|---|---|---|
| 0.5 | 77.49±0.18 | 76.80±0.22 | 77.05±0.36 |
| 1 | 77.25±0.25 | 76.35±0.21 | 76.90±0.37 |
| 5 | 81.37±0.21 | 74.76±0.15 | 80.16±0.36 |
| 10 | 76.84±0.84 | 74.14±0.16 | 84.76±0.25 |
| 20 | 57.36±0.10 | 75.58±0.18 | 86.73±0.20 |

Table 6: Test accuracy (in %) of adversarial regularization methods: AR, $WAR_{0-1}$, and WAR with different $\beta$ values on CIFAR-10 dataset with 40% noise level. The average accuracies and standard deviations over last 10 epochs are reported, and experiments are conducted only for one run.

### A.6    NOISY LABEL AT OBJECT SCALE FOR VAIHINGEN DATASET

We give the original and the used ground truth for Vaihingen dataset experiments.

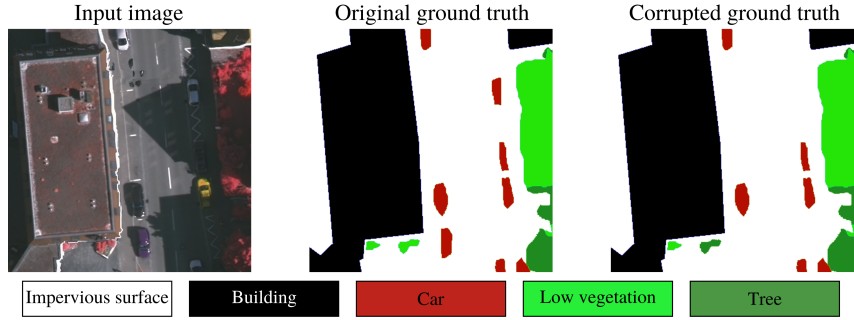

Figure 3: Comparison of the original ground truth and the corrupted ground truth.

### A.7    DISCUSSION ON CO-TEACHING+

In order to analysis low performance of Co-Teaching+(Yu et al., 2019), we conducted a series of experiments on CIFAR-10 with 40% noise level. Firstly, we used a similar architecture as the one used in their paper (pairflip 45% noise level) and reproduced their results (39%, 43% for Co-Teaching and Co-Teaching+ at 120 epochs). Next we conducted experiments with different settings to analysis sensitiveness of Co-Teaching+, and the results are shown in Table 7. When the pairflip noise and 2-layer CNN (one used in Co-Teaching+) is considered, Co-Teaching+ performs better than Co-Teaching, and it is opposite when the noise setting is changed. When the wide capacity model is considered, Co-Teaching+ is inferior in both noise settings. To understand this behaviour, we observed the pure ratio of the selected instances of both methods, and found that the pure ratio of selected instances decreases after few epochs of Co-Teaching+ loss (Co-Teaching+ uses a warm-up strategy with Co-Teaching method for 20 epochs). For example with CIFAR-10 at 40% noise, the pure ratio of selected instances is approximately around 57%, whereas in Co-Teaching it is around 70%. It is noted that in Yu et al. (2019) considered a unrealistic noise simulation (Pairflip), which flips the class labels with respect to the successive classes without considering class similarities, where as in our paper we have considered a class dependent noise, which flips the label according to the class similar classes (Patrini et al., 2017).

Table 7: Test accuracy of Co-Teaching and Co-Teaching+ with different model architectures and noise settings on CIFAR-10 with 40% noise level.

| Model | Noise | Co-Teaching | Co-Teaching+ |
|---|---|---|---|
| 2 layer CNN | pairflip | 47% | 52% |
| | class dependent noise | 56% | 54% |
| 9 layer CNN | pairflip | 78% | 64 % |
| | class dependent noise | 80% | 72% |

