# OpenReview forum: "Wasserstein Adversarial Regularization (WAR) on label noise"
_ICLR.cc/2020/Conference — Reject_

### Official Review · AnonReviewer2 · 2019-10-19
**Official Blind Review #2**

**Rating:** 3

**Review:**

This paper proposes to use the optimal transport distance to smooth the output of algorithms dealing with label noise. More specifically, the smoothness is on a controlled distance to the input instances, achieved by using the idea of adversarial learning (or robust learning). Generally, the paper is well-organized, with a good writing, and with sufficient experiments. However, I have major concerns about the technical contribution and the experiment parts.

1. It seems that the authors employed the existing adversarial learning techniques directly to the label noise problem. The technique contribution is quite limited. I think the challenging and interesting part should be about how to learn the metric C. Currently, the authors use the semantic distances between different class labels to set-up C. This intuitively makes sense but not convincing. We expect the authors to put more effort to study C, which may depend on the label noise rates or the geometric information of the instances. An in-depth study of C will make the technical contribution solid.

2. The experiment part looks sufficient but may be problematic. For example, in the subsection about Clothing1M, the authors directly cited the results from others while the experimental setting could be very different. It is not convincing to make the conclusion that the proposed method has the best performance.

**Experience Assessment:**

I have published in this field for several years.

**Review Assessment: Checking Correctness Of Derivations And Theory:**

I assessed the sensibility of the derivations and theory.

**Review Assessment: Checking Correctness Of Experiments:**

I assessed the sensibility of the experiments.

**Review Assessment: Thoroughness In Paper Reading:**

I read the paper thoroughly.

---

> ### Author Response · Authors · 2019-11-12
> **Thanks for your comments and analysis of our work**
>
> Dear reviewer,
>
>     We would like to thank you for your review which gives us the opportunity to clarify our contributions.
>
>     Regarding point 1 (limited technical contribution): in the paper, we introduce new regularization techniques dealing with an important problem involving a real-world challenge: label noise. We agree that both AR and WAR are based on adversarial training, which has been extensively studied in adversarial robustness or semi-supervised learning. Yet, we are the first (up to the best of our knowledge) to propose it for tackling label noise. We notably exhibit theoretical reasons to do so, in the form of Lemma 1.  Furthermore, our proposed WAR is clearly an original contribution based on the use of the Wasserstein distance: it extends the use of the Wasserstein distance on label space, rather than the data space, which provides a sound way of integrating pairwise relationships between classes, and regularize accordingly. The techniques are validated both from a theoretical point of view (by demonstrating that WAR provides robustness to label noise, Lemma 2) and in practice (by showing improvements on 5 commonly known datasets for classification and semantic segmentation). An additional advantage of the AR and WAR techniques is that, as regularizers, they could be added to any other standard architecture, including those designed specifically for label noise (such as Co-teaching). We also showed that no specific architectures were needed to achieve superior results when using (W)AR.
>
>
>     Regarding point 1 (learning the C matrix): we believe that obtaining the numerical improvements observed *without* learning the ground cost C is quite remarkable for the following reasons:
>
>     1) It shows that a ground cost based on no other prior knowledge than the class names is sufficient to take advantage of this regularization technique, especially when the noise quantity is unknown. This is not a trivial result and also a very important one, since the quantity and type of label noise are often very hard to estimate in real-world problems;
>
>     2) If additional prior knowledge is available, it can easily be embedded in the ground cost with minimal effort. While it might be possible to learn the ground cost in experimental settings where the noise is controlled, we argue that learning it, in a setting where *no ground truth* is known on the type of class switches that operate inside the data, could lead to overfitting, which would not be detectable. As stated above, in these cases the noise rates are generally unknown, and the geometrical information extracted from noisy labels might not be relevant to solve the problem. WAR operates without access to clean validation samples, so estimating C from clean samples is also not an option. For all those reasons, while we agree that studying ground cost estimation is a very interesting topic in itself, it is somehow out of the scope of this study, which is to establish the theoretical framework for adversarial regularization in fighting label noise.
>
>
>     Regarding point 2:  We have used, to the best of the details given in the related publications, the same experimental settings (model architecture, data augmentation, and optimizer) as the methods compared to our work. We also reproduced the results for the cross entropy in Table 3. During rebuttal we conducted additional experiments with the bootsoft method and our reproduced results is 68.75%, which is close to the one reported from the literature (68.94%). Based on the closeness of the results on two methods, we can confirm that the settings can be considered equivalent. It is noted that Forward method used a part of noisy and clean validation dataset for the estimation of noise transition matrix, which is usually not available in real-world noisy settings.  Lastly, we would like to stress that all the experiments on the 4 other datasets have been fully carried by us and show results in the same direction

---

> > ### Comment · AnonReviewer2 · 2019-11-14
> > **Others results on Clothing1m**
> >
> > I found that some other papers reported much higher rates on Clothing1m, e.g.,  72+ in the paper Joint Optimization Framework for Learning with Noisy Labels; and 73+ in Probabilistic End-to-end Noise Correction for Learning with Noisy Labels using a subset. It seems the proposed method cannot compete with them.

---

> > > ### Author Response · Authors · 2019-11-14
> > > **Fairness of comparisons**
> > >
> > > Dear reviewer, thank you for the feedback and pointing those recent papers. Indeed some higher performances are reported in this literature, but it is important to compare comparable results (a point also raised by the reviewer):
> > >
> > > - 72+ (joint optimization…): authors optimize hyperparameters of the model using a clean validation set, which we don’t. We believe our setting is more credible in a label noise setting: if it is possible to tune a model on clean samples, they could have been used for training. Apart from that, it is worth noting that they report test accuracy with respect to the best validation accuracy, while in our experiments we do not use validation samples at all and report the test accuracy at the end of training.
> > >
> > > - 73+ (noise correction): authors use a balanced custom made dataset, which makes the comparison with our results impossible.
> > >
> > > By modifying batch size and learning rate scheduling, we obtained 71.8% on the Clothing1M dataset. In the original submission, we decided not to report that result for reporting only fair comparisons using the same setting for all models . Please note that this last result, which is comparable in accuracy with those mentioned above, is still obtained in a completely unsupervised way, which we think is a strong result.
> > >
> > > The objective of the Clothing1M dataset experiment with our method is to show that our method also works well in the real-world noise case, and our results corroborate  this idea. Careful tuning of the hyper parameters, sometimes by monitoring the performance on a clean validation set or even the test set, might lead to better absolute performances, but we tried to the best of our capacity to preserve a strict and fair comparison  with existing methods.
> > >
> > > We hope that we have clarified all doubts.

---

### Official Review · AnonReviewer1 · 2019-10-23
**Official Blind Review #1**

**Rating:** 8

**Review:**


Update after rebuttal:
The rebuttal addresses my concerns. The point raised by reviewer 2 regarding other results on Clothing1M is concerning, but the authors' response is reasonable. Moreover, the proposed method is novel and could be complementary with other methods that achieve good results on Clothing1M. I think this paper would generate good discussion, and I recommend acceptance.

----------------------------------------------------------------------------------------
This paper presents a novel approach for dealing with asymmetric label noise. There are in fact two methods proposed, Adversarial Regularization (AR) and Wasserstein Adversarial Regularization (WAR), both of which derive from the intuition that model smoothing (in the sense of virtual adversarial training) acts as label smoothing when there are noisy labels. The authors prove two statements formalizing this notion and demonstrate the efficacy of AR and WAR in an extensive evaluation.

WAR is an iteration on AR that uses the optimal transport distance between categorical distributions to allow incorporating a cost matrix that encodes class similarities. The authors obtain these similarities from word embeddings of class names.

The evaluation demonstrates that the proposed WAR method achieves state-of-the-art results with the largest gains at high noise levels. In no case does it perform substantially worse than competing methods. Moreover, the AR method proposed as an intermediary to WAR also outperforms all prior work at non-zero noise levels, demonstrating the promise of the underlying approach.

The authors also evaluate on Clothing1M and evaluate on a semantic segmentation dataset for which they synthesize structured, realistic label noise. WAR outperforms prior work on these tasks. Moreover, the semantic segmentation task could act as a useful benchmark for future work.

For these reasons, I recommend acceptance.

Minor points:
Comparing the second and third columns in Figure 1, it looks like lambda has a large effect, so it would be good to know how much tuning lambda requires for the main tasks.

In equation 9, shouldn’t there be a beta in front of the entropy term? This does not break the result.

In equation 10, it looks like you’re multiplying both sides by (1 – epsilon), but you drop the (1 – epsilon) on the left side. This does not break the result.

**Experience Assessment:**

I have published one or two papers in this area.

**Review Assessment: Checking Correctness Of Derivations And Theory:**

I assessed the sensibility of the derivations and theory.

**Review Assessment: Checking Correctness Of Experiments:**

I assessed the sensibility of the experiments.

**Review Assessment: Thoroughness In Paper Reading:**

I read the paper at least twice and used my best judgement in assessing the paper.

---

> ### Author Response · Authors · 2019-11-12
> **Thanks for your comments and analysis of our work**
>
>  Dear reviewer,
>
>     Thank you for the constructive comments and for pointing out typos in Eq. (9) and Eq. (10). Those typos are fixed in the new version of the paper and we can confirm that they do not break the results.
>
>     We considered the entropic variant of the optimal transport loss because the solution is easily computable through the Sinkhorn algorithm with automatic differentiation, and can be parallelized on all the samples in a batch, due to the matrix/vectors product involved in its computation.   Using a big regularization parameter results in smoothing the loss, drives the solution away from the true OT solution but speeds up the Sinkhorn algorithm. As such, we recommend to use a small lambda to stay as close as possible to the true OT loss.  For all the experiments in the main task, we have used same regularization parameter ($\lambda$ = 0.05) with 20 sinkhorn iterations, showing that the regularization parameter is not too sensitive across datasets.

---

### Official Review · AnonReviewer3 · 2019-10-24
**Official Blind Review #3**

**Rating:** 6

**Review:**

===========
Summary:
This paper proposes a new regularization scheme inspired from (virtual) adversarial training to tackle the problem of learning with noisy labels. While based on the adversarial training (AR), it was found that AR does not directly transferable to deal with noisy labels. The author then proposed the Wasserstein version of AR replacing the KL with the Wasserstein distance and its approximate. This gives the proposed  Wasserstein Adversarial Regularization (WAR) which provide considerable robustness improvement on 5 datasets (both classification and segmentation). The correlation between WAR regularization and boundary smoothing is justified both theoretically and empirically with toy examples. The advantage of WAR regularization over existing methods is the flexibility to incorporate intra-class divergence, making it plausible against asymmetric label noise, which is more common in real-world datasets. The authors have done solid work in this paper. The experiment is complete, in terms of the scale, noise settings and comparison to existing works.

I recommend to accept this paper.

Minor suggestions:
It would be interesting to see the performance on the other common type of real-world noise: open-set label noise [1], and may be applied to adversarial training against adversarial examples.
The adversarial regularization was also used in a recent adversarial training paper [2]. A similar idea is adversarial logit pairing which is regularization on logits [3].


References:
[1] Iterative learning with noisy labels. CVPR 2018
[2] Theoretically principled trade-off between robustness and accuracy. ICML 2018
[3] Adversarial logit pairing. arXiv preprint arXiv:1803.06373 (2018).

**Experience Assessment:**

I have published in this field for several years.

**Review Assessment: Checking Correctness Of Derivations And Theory:**

I assessed the sensibility of the derivations and theory.

**Review Assessment: Checking Correctness Of Experiments:**

I carefully checked the experiments.

**Review Assessment: Thoroughness In Paper Reading:**

I read the paper thoroughly.

---

> ### Author Response · Authors · 2019-11-12
> **Thanks for your comments and analysis of our work**
>
> Dear reviewer,
> Thank you for your analysis of our work. Please find below answers to your minor comments:
>
>     Regarding 1 (open set label noise): Thank you for the constructive feedback, for the extra references and for pointing out an interesting real world noise setting. As suggested, we tried our WAR method for open set label noise. Following [1], we considered a simulated openset dataset composed of 60% of training data from CIFAR 10 and 40% from SVHN. We then tested the results on CIFAR-10. We privileged this more difficult setting  since the neural network can easily over-fit the samples from SVHN.  We used a similar neural network architecture and training procedure as detailed in [1], to directly compare our results. For WAR, we used similar hyper-parameters as used in the close-set noisy labels. We obtained the following performances:
>
>        * Wang et.al [1] = 77.73%
>         * WAR                 = 78.72%
>
>     The results shows that our method is also competitive in an openset noisy labels setting. Though our method does not have any specific functionality to tackle to openset label noise, the class geometry encoded in WAR is still effective in tackling openset noisy labels. Further, the accuracy of WAR could be improved by detection and reweighting of openset samples as in [1], or by using the idea of large loss samples, as in co-teaching based methods. Given the limited time for the rebuttal, we leave detailed experiments in those directions for future work.
>
>
>     Regarding 2: We agree that our method could also be applied to improve robustness against adversarial samples, especially since it would help regularizing more specific and critical class switches (stop sign VS other road sign for instance). But  we focus in this work only on the label noise and due to limited time, we leave the experiments and extension of WAR against adversarial samples for future work. The discussion related to [2] and [3] will be updated in the paper.
>
>
> Reference:
> [1] Iterative learning with noisy labels. CVPR 2018
> [2] Theoretically principled trade-off between robustness and accuracy. ICML 2018
> [3] Adversarial logit pairing. arXiv preprint arXiv:1803.06373 (2018).

---

### Decision · Program_Chairs · 2019-12-19

**Decision:**

Reject

**Comment:**

This article proposes a regularisation scheme to learn classifiers that take into account similarity of labels, and presents a series of experiments. The reviewers found the approach plausible, the paper well written, and the experiments sufficient. At the same time, they expressed concerns, mentioning that the technical contribution is limited (in particular, the Wasserstein distance has been used before in estimation of conditional distributions and in multi-label learning), and that it would be important to put more efforts into learning the metric. The author responses clarified a few points and agreed that learning the metric is an interesting problem. There were also concerns about the competitiveness of the approach, which were addressed in part in the authors' responses, albeit not fully convincing all of the reviewers. This article proposes an interesting technique for a relevant type of problems, and demonstrates that it can be competitive with extensive experiments. ``Although this is a reasonably good article, it is not good enough, given the very high acceptance bar for this year's ICLR.